# Immune Transcriptome Study of Human Nucleated Erythroid Cells from Different Tissues by Single-Cell RNA-Sequencing

**DOI:** 10.3390/cells11223537

**Published:** 2022-11-09

**Authors:** Roman Perik-Zavodskii, Olga Perik-Zavodskaia, Julia Shevchenko, Vera Denisova, Saleh Alrhmoun, Marina Volynets, Valeriy Tereshchenko, Konstantin Zaitsev, Sergey Sennikov

**Affiliations:** 1Laboratory of Molecular Immunology, Federal State Budgetary Scientific Institution Research Institute of Fundamental and Clinical Immunology, 630099 Novosibirsk, Russia; 2Clinic of Immunopathology, Federal State Budgetary Scientific Institution Research Institute of Fundamental and Clinical Immunology, 630047 Novosibirsk, Russia; 3Federal State Budgetary Scientific Institution “Siberian Federal Research and Clinical Center of the Federal Medicobiological Agency”, 634009 Tomsk, Russia

**Keywords:** nucleated erythroid cells, red blood cells, bone marrow, foetal liver, umbilical cord blood, single-cell RNA sequencing, scRNA-seq

## Abstract

Nucleated erythroid cells (NECs) are the precursors of erythrocytes. They can be found in various hematopoietic tissues or in the blood. Recently, they have been shown to be active players in immunosuppression through the synthesis of arginase-2 and reactive oxygen species. In this work, we studied NECs in adult bone marrow, umbilical cord blood, and foetal liver parenchyma using single-cell RNA sequencing and found that: (1) all studied NECs expressed the same set of genes, which was enriched in “GO biological process” immunity-related terms; (2) early and late NECs had differential expression of the genes associated with immunosuppression, cell cycle progression, apoptosis, and glycolysis; (3) NECs from different tissues of origin had differential expression of the genes associated with immunosuppression.

## 1. Introduction

Nucleated erythroid cells (NECs) are the precursors of erythrocytes. NECs consist of proerythroblasts, basophilic erythroblasts, polychromatophilic erythroblasts, and oxyphilic erythroblasts and are defined by the presence of CD235a (glycophorin A) on their cell surface [1]. There are many known processes that are carried out by NECs, such as the transport of oxygen and molecules. In recent years, a new function of NECs has been discovered—the function of immunoregulation. This has been shown by Delyea et al. [2]: erythroid cells can cause immunosuppression due to the production of arginase-2 and reactive oxygen species [3]. Other works have also shown that NECs can express mRNA and produce multiple cytokine proteins [4,5,6,7,8]. In this work, we took advantage of the single-cell mRNA sequencing capabilities of the Becton Dickinson (BD) Rhapsody to learn more about the immunoregulatory potential of NECs using the “Immune Response” gene panel.

## 2. Materials and Methods

### 2.1. Study Population

Adult bone marrow donors were healthy 24–26-year-old males (*n* = 3). Foetal thymus parenchyma cells (*n* = 3) and foetal liver parenchyma cells (*n* = 3) samples were collected after an abortion at 20–22 weeks of pregnancy; sex is unknown. Umbilical cord blood (*n* = 3) was collected after childbirth; sex is unknown. Bone marrow harvest was approved by a local ethics committee. Donors gave written informed consent.

### 2.2. Study Interventions

We performed punctures and bone marrow aspirations (*n* = 3).

### 2.3. Participant Safety

Three days prior to the bone marrow donation procedure, bone marrow donors underwent a medical examination, including complete blood count, electrocardiogram, and qPCR for COVID-19. We observed the donors for 60 min after the bone marrow harvesting procedure.

### 2.4. Cell Isolation

We collected bone marrow (up to 5 mL) in EDTA-containing tubes. We thawed liquid nitrogen stored cord blood, foetal thymus parenchyma, and foetal liver parenchyma samples (up to 1.5 mL) in a 37 °C water bath and then washed them with a 6 mL mixture containing 5 mL of complete RPMI 1640 cell culture medium and 1 mL FBS. We isolated bone marrow, umbilical cord blood, foetal thymus parenchyma, and foetal liver parenchyma mononuclear cells by density gradient centrifugation (Ficoll-Paque 1.077 g/mL) at 266 RCF for 30 min, while simultaneously removing erythrocytes.

### 2.5. Cell Sorting

We performed magnetic sorting of bone marrow, umbilical cord blood, foetal thymus parenchyma, and foetal liver parenchyma mononuclear cells using a magnetic stand and magnet (Miltenyi Biotec, Bergisch Gladbach, Germany, 130-042-102) and anti-CD235a MicroBeads (Miltenyi Biotec, Bergisch Gladbach, Germany, 130-050-501), in accordance with the protocols of the manufacturer.

### 2.6. Viability Staining

We measured magnetically sorted cells’ viability on a Countess 3 Automated Cell Counter (Thermo Fisher Scientific, Waltham, Massachusetts, USA), in accordance with the protocols of the manufacturer, using Trypan Blue. Trypan blue staining showed >95% viability for the bone marrow CD235a^+^ samples, >65% viability for the foetal thymus parenchyma CD235a^+^ samples, >85% viability for the foetal liver parenchyma CD235a^+^ samples, and >95% viability for the cord blood CD235a^+^ samples.

### 2.7. Sample Tag Barcoding

After cell sorting, we incubated individual NECs subsets with Sample Tag antibodies for 20 min at room temperature. After three washing cycles, cells were counted, equal numbers of CD235a^+^-sorted cells were pooled from adult bone marrow cells (*n* = 3, male donors), umbilical cord blood cells (*n* = 3, gender unknown), foetal liver parenchyma cells (*n* = 3, gender unknown), and foetal liver parenchyma cells (*n* = 3, gender unknown) together and resuspended in cold sample buffer to a final concentration of 40 cells/µL and a volume of 620 µL for loading onto a BD Rhapsody Cartridge. The quality of cell loading into the cartridge was assessed using the InCell Analyzer 2000.

### 2.8. cDNA Library Preparation and Sequencing

We performed single-cell capture and cDNA library preparation using the BD Rhapsody Express Single-Cell Analysis System (BD Biosciences, Franklin Lakes, NJ, USA), in accordance with the instructions of the manufacturer. Briefly, we amplified cDNA (10 cycles of PCR) using the Human Immune Response Primer Panel (BD Biosciences, Franklin Lakes, NJ, USA, 633750), containing 399 primer pairs and targeting 397 different genes. We purified the resulting PCR1 products using AMPure XP magnetic beads (Beckman Coulter, Brea, CA, USA) and separated the respective mRNA and Sample Tag products on the basis of amplicon size. We further amplified the purified mRNA and Sample Tag PCR1 products (10 cycles of semi-nested PCR) and purified the resulting PCR2 products by size selection. We assessed the concentration by Qubit (High-Sensitivity dsDNA Kit, Thermo Fisher Scientific, Waltham, MA, USA). We normalized the final products to 2.5 ng/μL for the mRNA library and 1.0 ng/μL for the Sample Tag library and performed a final round of amplification (6 cycles of PCR for mRNA library and 8 cycles of PCR for Sample Tag library) using indexes for Illumina sequencer to prepare the final libraries. We quantified the final libraries using Qubit and Agilent BioAnalyzer 2100 (Agilent, Santa Clara, CA, USA) and pooled them (~94%/6% mRNA/Sample Tag ratio, estimated 2300 (mRNA) and 130 (Sample Tag) reads/cell) to achieve a final concentration of 5 nM. The final pooled libraries were outsourced for sequencing to Institute of Cytology and Genetics (Novosibirsk), where they were enriched with 20% PhiX control DNA to increase sequence complexity and sequenced (75 bp paired-end, 130 million clusters) on a NextSeq 550 sequencer (Illumina, San Diego, CA, USA). 

### 2.9. Data Processing

We processed the FASTQ files obtained from sequencing using the BD Rhapsody pipeline v1.9 (BD Biosciences). Initially, the pipeline removed read pairs with low quality based on their read length, mean base quality score, and highest single-nucleotide frequency; analysed remaining high-quality R1 reads in order to identify cell label and unique molecular identifier (UMI) sequences; and aligned the remaining high-quality R2 reads to the reference panel sequences (mRNA) using Bowtie2. The pipeline then collapsed reads with the same cell label, the same UMI sequence, and the same gene into a single molecule. The pipeline adjusted the obtained counts by error-correction algorithms, namely recursive substitution error correction (RSEC) and distribution-based error correction (DBEC), in order to correct for sequencing and PCR errors. Then, the pipeline estimated cell counts using the second derivative analysis to filter out noise cell labels. The pipeline observed one inflexion point and considered cell labels after that to be noise labels. Then, the pipeline used molecular barcoded oligo-conjugated antibodies (single-cell multiplexing kit; BD Biosciences) to demultiplex the samples and filter out some of the multiplets. The pipeline called 3906 cells for CD235a^+^ adult bone marrow cells sample, 1689 cells for CD235a^+^ cord blood cells sample, 37 cells for CD235a^+^ foetal thymus parenchyma cells sample, and 3774 cells for CD235a^+^ foetal liver parenchyma cells sample. We manually discarded CD235a^+^ foetal thymus parenchyma cells sample due to the low number of cells called. Paragraphs 2.4, 2.5, and 2.7–2.9 are depicted in Figure 1.

### 2.10. Data QC and Analysis

We used either SeqGeq 1.6 (FlowJo, Ashland, OR, USA) or Seurat 4.0 [9] for quality control, data analysis, and plotting.

In SeqGeq, we performed counts per million normalization of all single cells in the data sets; gated singlets from all cells; gated ALAS2^+^ and SLC25A37^+^ cells from the singlets to only include NECs in the downstream analysis (both genes were shown to have erythroid single-cell specificity in Protein Atlas database: *ALAS2* (5’-aminolevulinate synthase 2)—https://www.proteinatlas.org/ENSG00000158578-ALAS2 (accessed on 17 July 2022), *SLC25A37* (Solute carrier family 25 member 37)—https://www.proteinatlas.org/ENSG00000147454-SLC25A37 (accessed on 17 July 2022)), demultiplexed biological groups using the Sample Tag-one-hot-encoded information, and split each NECs’ biological group data set into early and late NECs data sets using a known NECs’ maturation marker, *ITGA4* (*ITGA4* encodes for the α4-integrin, a surface marker of NECs with a surface expression that continuously decreases as NECs mature [10]) (Figure 2). It also helped to balance the data sets, as NECs from different tissues had different percentages of early and late NECs (adult bone marrow NECs had 57.4% early NECs and 42.6% late NECs, cord blood NECs had 24.3% early NECs and 75.7% late NECs, and foetal liver parenchyma NECs had 38.7% early NECs and 61.3% late NECs). 

We used Uniform Manifold Approximation and Projection (UMAP) for dimensionality reduction. We subjected a set of genes expressed in a majority of NECs to gene set enrichment analysis using “GO biological process 2021” gene set using GSEApy (https://github.com/zqfang/GSEApy, accessed on 15 October 2022). We performed every possible pairwise differential gene expression analysis between early and late adult bone marrow NECs, early and late cord blood NECs, and early and late foetal liver parenchyma NECs using Mann–Whitney U test with FDR *p*-value correction for multiple testing in SeqGeq, and we created differential gene expression volcano plots in GraphPad Prism 9.4. 

As for the Seurat 4.0, we manually selected ALAS2^+^ and SLC25A37^+^ cells from all cells and split each NECs’ biological group data sets into early (*ITGA4* positive) and late (*ITGA4* negative) via Jupiter Notebooks and Pandas. Then, we performed doublet QC and SCTransform of each bio group data set independently, integrated the independent data sets using *FindIntegrationAnchors* and *IntegrateData* methods, and created Ridge and Dot plots of differentially expressed genes using Seurat 4.0 in R Studio.

## 3. Results

### 3.1. Dimensionality Reduction

We used the UMAP dimensionality reduction method in order to assess the gene expression variation of the studied NECs. UMAP output three cell clusters, and each UMAP-output cluster matched a single tissue of the NECs’ origin (Figure 1). 

We found out that the NECs from the foetal liver parenchyma had unique *CD8A* and *IL33* gene expression compared with the other NECs; the NECs from the cord blood had unique *FN1, CD22*, and *STAT4* gene expression compared with the other NECs; and the NECs from the adult bone marrow had no unique gene expression compared with the other NECs. It should be noted that only a small number of cells (5–30 cells) had expression of these genes.

### 3.2. NECs’ Gene Set Analysis

We found that a majority of the NECs from the adult bone marrow, the cord blood, and the foetal liver parenchyma have gene expression of the same set of genes: *ALAS2*, *ARG1*, *AURKB*, *BAX*, *BTG1*, *C10orf54*, *CD36*, *CD44*, *CNOT2*, *CXCL5*, *CXCL8*, *DEFA3*, *DUSP1*, *FAS*, *FOSB*, *FTH1*, *GAPDH*, *HMMR*, *IGBP1*, *IKZF1*, *IL15RA*, *IL23R*, *ITGA4*, *ITGAE*, *KCNE3*, *KIAA0101*, *LAMP1*, *LAP3*, *LGALS3*, *LGALS9*, *MCM2*, *MCM4, MKI67*, *MYC*, *NAMPT, NINJ2*, *PASK*, *PCNA*, *PDIA6*, *PIK3AP1*, *PTTG2*, *RORA*, *RPN2, S100A9*, *S100A10*, *S100A12*, *SLC25A37*, *SNCA*, *TOP2A*, *TSPAN32*, *TYMS*, *UBE2C*, and *VEGFA*. 

A minority of the NECs expressed other genes, such as *DEFA4, IGHE (secreted),* MHC class II genes (yet no single cell had expression of both MHC class II chains), a cluster of differentiation genes, as well as a plethora of cytokine and chemokine genes (see Appendix A).

We then performed gene set enrichment analysis using the GO Biological Process for this set of genes and observed an enrichment in several immunity-related terms (Table 1). The top 10 results are depicted in Figure 2. The top result was the “antimicrobial humoral immune response mediated by antimicrobial peptide” term.

Several genes were enriched cell percentage expressing-wise in most of the studied NECs: *ALAS2*, *ARG1*, *CD36*, *CD44*, *CXCL5*, *ITGA4*, *LGALS3*, and *SLC25A37* (Figure 3).

### 3.3. Early and Late NECs have Differentially Expressed Genes

We performed pairwise differential gene expression (DE) analyses between the early and late NECs for every studied tissue of the NECs’ origin, and we considered a *q*-value < 0.05 and a log2 (fold change) > ±0.847 significant (Figure 4a–c).

We observed that: (1) late NECs from every tissue of origin have higher *ARG1* gene expression compared with early NECs; (2) early NECs from every tissue of origin have higher *CD36* gene expression compared with late NECs; (3) early NECs from every tissue of origin have higher cell cycle S-phase (Figure 5a) and M-phase (Figure 5b) genes’ gene expression compared with late NECs; (4) early NECs from every tissue of origin have higher gene expression of the pro-apoptotic gene *BAX* compared with late NECs; (5) early NECs from every tissue of origin have higher gene expression of the glycolysis gene *GAPDH* compared with late NECs. 

We obtained the same results (inverse correlation of *ARG1* and *CD36* gene expression) using Monocle for the adult bone marrow and cord blood NECs (see Appendix A) and via UMAP dimensionality reduction and clustering—the majority of *ARG1* gene expression was detected in the *CD36* and *ITGA4* low/dim clusters (Figure 6a–c).

We then performed pairwise early versus early NECs (Figure 7a–c) and late versus late NECs (Figure 8a–c) differential gene expression analyses.

We observed that: (1) **early** cord blood NECs had higher *ARG1* gene expression compared with either **early** adult bone marrow NECs or **early** foetal liver parenchyma NECs; (2) **late** cord blood NECs had higher *ARG1* gene expression compared with either **late** adult bone marrow NECs or **late** foetal liver parenchyma NECs; (3) **late** cord blood NECs had lower *LGALS3* gene expression compared with either **late** adult bone marrow NECs or **late** foetal liver parenchyma NECs. 

We have summarized the expression data for the most common differentially expressed immunity-related genes in Figure 9.

## 4. Discussion

In this work, we found transcriptomic differences between the adult bone marrow, cord blood, and foetal liver parenchyma NECs. Our results showed that: (1) all studied NECs expressed the same set of genes, which was enriched in “GO biological process” immunity-related terms; (2) early and late NECs had differential expression of the genes associated with immunosuppression, cell cycle progression, apoptosis, and glycolysis; (3) NECs from different tissues of origin had differential expression of the genes associated with immunosuppression.

It has been well-established that NECs have immunosuppressive properties [2,11,12,13,14,15,16] and play a big role in the pathogenesis of many important infectious diseases, such as COVID-19 [17]. Our data allow us to make several assumptions about the NECs’ immunosuppressive potential. We can propose that NECs have a contact mechanism of immunosuppression through Galectin-3 [18] and Galectin-9 [19]. Galectin gene expression was previously found in the CECs of mice [8]. Our analyses also show that the early NECs from every studied tissue have lower ARG1 gene expression compared to the late NECs from the same tissues.

We can also propose that late cord blood NECs can be more immunosuppressive using the enzymatic mechanism of immunosuppression through arginase-1, as cord blood NECs had the highest *ARG1* expression among the studied NECs and less immunosuppressive using the contact mechanism, as they have the lowest *LGALS3* gene expression.

NECs could also possibly have angiogenic properties, as they have *VEGFA* gene expression [20]. We found gene expression of CXCL5 and CXCL8 in all studied NECs. It is known that CXCL5 and CXCL8 facilitate neutrophil chemotaxis [21]. Thus, it is possible that NECs can play a role in neutrophil recruitment. In general, early NECs show a trend towards higher expression of the CXCL5 gene compared to late NECs from the same tissues.

We also found alpha-defensin 3 gene expression, a gene encoding for an antimicrobial, antifungal and antiviral peptide [22]. This possibly makes NECs players in innate immunity.

Co-expression of antimicrobial gene *DEFA3* and immunosuppression-inducing genes *ARG1* and *LGALS3* could allow NECs to both combat invading pathogens, while insuring the overall local immunosuppression, which can be helpful at the developmental stages of ontogenesis, such as in a newborn’s gut.

It has been previously shown that NECs have cytokine gene expression and protein production [3,4,5,6,7,8]. Apart from *CXCL5* and *VEGFA* genes, only a minuscule number of NECs had gene expression of any other cytokine.

We found that CD36 is up-regulated in early NECs compared with late NECs, which is in accord with the previous key study [10].

Up-regulation of the pro-apoptotic gene *BAX* in early NECs compared to late NECs could mean that early NECs are more susceptible to apoptosis-inducing factors.

Up-regulation of the S-phase and M-phase genes in early NECs compared to late NECs could indicate an active mitosis at this stage and is in concordance with previous studies [23,24].

Up-regulation of the glycolysis gene *GAPDH* in early NECs compared to late NECs could mean that early NECs utilize glucose faster. This is in concordance with the up-regulation of the cell cycle progression genes, as such processes involve a lot of ATP-dependent biosynthesis.

We found that NECs express only two cytokine receptors (IL15RA and IL23R) out of all present in the panel (https://www.bd.com/documents/specifications/genomics/GMX_BD-Rhapsody-immune-response-human-panel_SP_EN.pdf, accessed on 15 October 2022). Out of these two, only IL15RA can bind its cytokine (IL-15) without any additional receptor subunits. Previously, it was shown that IL-15 can suppress erythropoiesis [25]. Early adult bone marrow NECs have higher *IL15RA* gene expression compared to late adult bone marrow NECs, and, thus, could be more susceptible to IL-15-caused suppression of erythropoiesis.

Quite strange for us was the discovery of the expression of the *IGHE (secreted)* gene. The expression of the light chain genes was not detected. Possibly, this may mean that a small number of NECs can produce IgE heavy chain paraproteins. It is difficult to assess the involvement of such potential paraproteins in immunity at this stage.

While some NECs had either alpha or beta chain paralogs of class II MHC molecules, no cell had both. This allows us to suggest that NECs cannot present any peptide in an MHC class II-dependent pathway.

In conclusion, it can be noted that the expression of many of the genes involved in immunoregulation and homing changes quantitively and qualitatively in erythroid cells during their differentiation. Along with the expression of similar genes in the erythroid cells of the bone marrow, umbilical cord blood, and foetal liver parenchyma, these erythroid cell populations have differences in gene expression levels both among themselves and within each tissue, which may indicate the presence of different subpopulations of erythroid cells. Very rare erythroid cell subpopulations are also observed, which express many unusual genes such as MHC class II genes, an uncharacteristic differentiation gene cluster, IGHE (secreted), and many cytokine and chemokine genes.

The limitations of this study were: (1) we only used ITGA4 to separate NECs into early and late because the second important marker SLC4A1 (band3) was absent from the gene expression panel, (2) we used gene expression data to separate the NECs into early and late, not the surface protein expression data that is commonly used.

Our future research will focus on validating gene expression data at the protein level: (1) chemokine secretion profiling, (2) CXCL5 based migration assays, (3) profiling of NEC antimicrobial activity with Defensin Alpha 3 and Defensin Alpha 4, (4) profiling of Galectin-3 and Galectin-9 on NECs’ cell surface, and (5) ARG1 Western blot and arginine-depletion assays for NECs from different tissues.

For all future nucleated erythroid cell researchers who plan to use scRNA-seq, we recommend: (1) sorting the NECs as CD71 and CD235a double-positive cells using a FACS machine; (2) usage of anti-CD71, anti-CD235a, anti-band3, and anti-CD49d antibodies with nucleotide barcodes for sequencing (TotalSeq, AbSeq, or any other), as they will allow for the best separation of the NECs at each stage of their differentiation, and it is possible to co-stain NECs with antibodies for scRNA-seq and FACS; (3) sorting the NECs at each stage of differentiation from CD71 and CD235a double-positive cells using band3 and CD49d for scRNA-seq validation purposes.

## Data Availability

The single-cell gene expression data were deposited into the Gene Expression Omnibus (GEO), with the accession code GSE199230.

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
