# Peer review of "Immune Transcriptome Study of Human Nucleated Erythroid Cells from Different Tissues by Single-Cell RNA-Sequencing"

_cells, 2022, doi:10.3390/cells11223537_

Round 1

Reviewer 1 Report (Previous Reviewer 3)

I thank the authors for addressing all the points, particularly the distinction between the early and late NECs. 

Author Response

Dead Reviewer, thank You, once again, for Your review!

Reviewer 2 Report (New Reviewer)

The role of nucleated erythroid cells as cellular mediators of the immune response, in particular in fish, has emerged as an interesting research subject. During the past decade, a number of biological processes relevant to immunity had been described for fish RBCs: (i) pathogen recognition; (ii) clearance of pathogens by means of binding microbial immune complexes and (iii) production of cytokines or specific signalling molecules in response to pathogens. Some researches are taken in this same direction in mammalians however it’s essential keep in mind that the immune system of fish is peculiarly dissimilar to that of higher vertebrates.

Here the authors explore an immune signature in NECs from BM, CB and FL by scRNASeq technology. Despite the interesting idea behind the paper, several points are imprecise and unclear and more investigations and considerations are needed.

Compared to bulk RNA-seq, scRNA-seq produces nosier and more variable data. Due to the high technical noise, QC is crucial for identifying and removing the low-quality scRNA-seq data to get robust and reproducible results. Several questions still open concerning this point. Why choose ALAS2 and SLC25A37 for normalization? Both are from mitochondria and it’s known that mitochondrial RNAs are retained for broken cells whereas cytoplasmic RNAs are usually lost. Several other and better characterized erythroid specific genes from cytoplasm or nucleus could be used to be sure that only NEC cells are included. Moreover, in the fig2 we can see that ALAS2 and SLC25A37 expression are not the same in the different groups. What means? This can have an impact in the final results?

To correctly interpret the results from scRNA-seq data, normalization is an essential step to get the signal of interest by adjusting unwanted biases resulted from capture efficiency, sequencing depth, dropouts, and other technical effects. Within-sample normalization is described but how the authors have taken in account the between-sample normalization? No information concerning this point is detailed in MM.

A key goal of scRNA-seq data analysis is to identify cell subpopulations within a certain tissue to unravel the heterogeneity of cells. Why the authors imposed a split in two population and didn’t use rather an unsupervised clustering approach? Moreover, ITGA4 is a questionable choice. In the literature is well described that to discriminate the erythroblasts stages the combination of CD49d and Band3 is necessary, ITGA4 only is not sufficient. It’s crystal clear that the early group is a very heterogenous group compare to the late one. In the fig 2 for every early groups we have 4 different expression levels of ITGA4: 4 different populations?

The authors showed the % of cells in each group but not the number. The most relevant differences are shown in  CB but attention the number of cells in this group is lower compared to the other two (half of cells) and moreover only 24% of cells are in the early group: 400 cells are not really enough to support valid conclusion!

Attention to another potential bias introduced in the study: proE are CD235a low/dim. Are you sure that magnetic sort is able to purify it? Why no FACS plots are shown to support the purity of the sorts and check that in all analyzed tissues all erythroblasts stages are present? Why GPA and not CD71 as in Delya et al is used?

Why to talk to thymus if not enough of cells to show results???

The cells during erythropoiesis exhibit a continuous spectrum of states and involve transitions between different erythroblasts states. Such dynamic processes within a portion of cells can be computationally modeled by reconstructing the cell trajectory and pseudotime based on scRNA-seq data. Do the authors have explored this way to better dissect heteorgeneity and identified the "right" populations during erythropoiesis? Or at least to check that their "imposed" populations show the right time order?

How many genes are identified by the scRNASeq approach? Only the genes listed in the 3.2 paragraph? What about the key regulators’ gene of erythropoiesis as GATA1, KLF1 etc?

Quite strange results concerning CD36 expression, it was shown several times by FC that all erythroblasts express CD36 (of course at the different levels), how to explain the negative peak in the fig2?

It’s regrettable the lack of details for table 1 concerning the gene name included in the gene sets.

Concerning the volcano plots of DE genes is quite strange that we have only one gene and always the same upregulated in the late populations. Experimental bias? How the authors have taken in account the well known decrease of transcriptional activities during erythropoiesis?

The idea behind the paper is interesting but the authors show mainly preliminary results. Only scRNASeq approach is used in the paper, the target genes are not checked with other techniques, why? qPCR in minibulk sorted populations? Protein expression by FACS to gate the appropriate populations? No functional assays. It’s essential to provide other experimental evidences to support their hypothesis.

Author Response

Dear Reviewer, thank you for your review! We tried our best to clarify and/or prove every point of Your interest! All the details are in the attached document!

Reviewer 3 Report (New Reviewer)

I thought that the paper was carefully done, demonstrating important changes between fetal and adult nucleated red cells transcriptome and some of the observations are novel and will provide the important impetus for future studies.

It would be helpful to know if these transcripts are productively translated into proteins.

The access to fetal tissue is likely unique and these data are believable and should be valuable to other investigators.

Author Response

Dead Reviewer, thank You for Your work and for the appreciation of our Manuscript! 

Round 2

Reviewer 2 Report (New Reviewer)

First of all, I thank the authors for addressing all the points of my previous report and for clarifying their research methodology. Deep explanations concerning the key stages of scRNA-Seq analysis consolidate the final paper message. However, some points still remain approximate. Of course, ITGA4 expression can be use to split early and late erythroid cells, but during erythropoiesis it’s known that ProE and E and LBaso have different functions, the same for Poly and Ortho cells; maybe it will be the same for their immune functions. I suggest the author to deeply explore this possibility in their next studies and to pay closer attention to ProE population that, despite its low percentage, has a higher degree of cellular plasticity compared to other erythroblasts populations.

I understand that the paper focus on erythroid cells as immune cells but I find it regrettable the total absence of master erythroid regulators in the panel and FACS analysis: to have the “good” positive controls in every experiment is never superfluous.

The paper message is quite innovative and exciting but the big problem is that all the conclusions are based on only one technique. scRNAseq is a powerful tool but as all experimental techniques is not perfect and its results need to be confirmed with other techniques. Moreover, several paper in erythroid field have already shown not a totally perfect correlation between mRNA and protein levels in erythroid cells. I strongly suggest to the authors to integrate some simple experiments (ex qPCR, WB etc) to check at least one or two targets between the most common differentially expressed immunity-related genes in the different population. 

In the discussion, it's important to point out some important limitation of the study: the mRNA identified may not be readily led to corresponding protein expresion and functional phenotypes (at least based on this paper version).

Author Response

Dear Reviewer, 

We were glad to clarify every point of our methodology!

Exclusion of CFU-E, BFU-E and none/dim-GYPA proE from this study is indeed a limitation, a regrettable, yet the only one that was and still possible in our lab due to the absence of a FACS machine. We intend to get one as soon as possible, as it will be the key to all the planned validation (qPCR, Western Blot, Arginine depletion by ARG1 and Trans-Well migration assays) of the transcriptomic data in our future studies.

We tried our best to outline every limitation of our study in the Discussion section:

"Limitations of this study were: 1) we only used ITGA4 to separate NECs into early and late because the second important marker SLC4A1 (band3) was absent from the gene expression panel, 2) we used gene expression data to separate NECs into early and late, and not surface protein expression data that is commonly used.

Our future research will focus on validating gene expression data at the protein level: 1) chemokine secretion profiling, 2) CXCL5 based migration assays, 3) profiling of NEC antimicrobial activity with Defensin Alpha 3 and Defensin Alpha 4, 4) profiling of Galectin-3 and Galectin-9 on NECs’ cell surface, 5) ARG1 western blot and arginine depletion assays for NECs from different tissues."

We also added recommendation for every future scientist that would like to study nucleated erythroid cells via scRNA-seq:

"For all future nucleated erythroid cell researchers who plan to use scRNA-seq, we recommend: 1) sorting NECs as CD71, CD235a double positive cells using a FACS machine, 2) usage of anti-CD71, anti-CD235a, anti-band3 and anti-CD49d antibodies with nucleotide barcodes for sequencing (TotalSeq, AbSeq or any other), as they will allow the best separation of NECs at each stage of their differentiation, it is possible to co-stain NECs with antibodies for scRNA-seq and FACS, 3) sort NECs at each stage of differentiation from CD71, CD235a double positive cells using band3 and CD49d for the scRNA-seq validation purposes."

This manuscript is a resubmission of an earlier submission. The following is a list of the peer review reports and author responses from that submission.

Round 1

Reviewer 1 Report

I am sorry to provide the following review but stand with my evaluation. In general the writing (grammar), presentation and conclusion of this manuscript is severely lacking compared to scientific standards. Sometimes there are even parts where the authors forgot to provide crucial information, e.g. “Supplementary Materials: The following supporting information can be downloaded at:”, this is not acceptable. There is no hypothesis, the aims of the study are unclear, the introduction is just a set of sentences without coherence, similar for the abstract. The results are at best descriptive with again no coherence and figures are not providing all the details (some only show some points without knowing what they represent (the volcano plots, we need the expression data for all genes as a bare minimum). No tables of gene expression are provided, supplementary data is rudimentary and not referred to. No flow cytometry data to show the purity of the populations for all the samples. In addition, the authors fail to indicate other studies that could have been datamined to specifically look at the couple of genes they are interested in. The discussion is not providing an embedding in existing literature and only provides some loose correlations and suggestion without any foundation. I do have a array of other specific points but reading the present state of the manuscript, I do not feel the need for further in depth review.

Reviewer 2 Report

Minor check required:

1) Full name of BFU-E, CFU-E, ALAS2 in Introduction and ECG in materials and methods and UMAP in Results need to be mentioned. (line 35, line 41, line 56, line 153)

2) In materials and Methods, proper abbreviation must be used. For example, y.o. (line 47) is not needed to be abbreviated. 

3) In materials and Methods, minutes or min of time unit are mixed up. They should be used collectively. (line 57, line 65, line 87)  

4) Typo for volume sample and q-value is needed to be changed as decimal point. (line 61, line 104, line 183)

5) Typo for degree Celsius: 37C (line 61)

Reviewer 3 Report

In the manuscript submitted, Perik-Zavodskii and colleagues analyzed the gene set of nucleated erythroid cells (NECs) taken from adult bone marrow, cord blood, and fetal liver. The results confirm the gene signature independently of the cell source is related to Immunity. Previously, NECs were already considered immunosuppressive agents because of the arginase-1 synthesis. Although the work is exciting at the scientific and technical levels, there are several issues to address before acceptance. The study's main limitation is that the authors considered the cells a homogenous population. NECs are a mix of cells at distinct differentiation stages (pro, basophilic, polychromatic, and orthochromatic erythroblasts). Considering that the % of each population depends on the donor and the source, I strongly recommend performing the experiments in sorted populations (at least two populations like pro-baso vs. poly-ortho). Other points to address are:

1-   It is known that mRNA expression is not always related to protein levels in human erythropoiesis (doi:10.1016/j.celrep.2016.06.085). Please, discuss and, if possible, show the protein levels of some immunosuppression-related genes.

2-   CD36 is a good marker of erythropoiesis, but not for these populations. CD36 mRNA levels are stable between basophilic and polychromatic cells and decrease in orthochromatic cells (doi: 10.1182/blood-2014-01-548305). Thus, the interpretation of the results could be wrong. Please reconsider using CD36 as a marker of erythroid differentiation in terminal erythropoiesis. The best candidates are SLC4A1 (Band3) or SLC2A1 (GLUT1).

 3-   The expression of arginase 1 is quite late in the erythroid differentiation and then is conserved in mature red blood cells. For example, the inverse correlation of CD36 and ARG1 can be found in orthochromatic erythroid progenitors and not in less differentiated cells. What’s happened with arginase 2?

 4-   The discussion is relatively short and should go deeper into your results.

 5-   Section 2.2: Please, explain in more detail.